# Verteporfin-Loaded Lipid Nanoparticles Improve Ovarian Cancer Photodynamic Therapy In Vitro and In Vivo

**DOI:** 10.3390/cancers11111760

**Published:** 2019-11-08

**Authors:** Thierry Michy, Thibault Massias, Claire Bernard, Laetitia Vanwonterghem, Maxime Henry, Mélanie Guidetti, Guy Royal, Jean-Luc Coll, Isabelle Texier, Véronique Josserand, Amandine Hurbin

**Affiliations:** 1Institute for Advanced Biosciences, Institut National de la Santé Et de la Recherche Médicale INSERM U1209, Centre National de la Recherche Scientifique CNRS UMR5309, Université Grenoble Alpes, F-38000 Grenoble, France; 2Centre Hospitalier Universitaire CHU Grenoble Alpes, Université Grenoble Alpes, F-38000 Grenoble, France; 3Centre National de la Recherche Scientifique CNRS UMR5250, Département de Chimie Moléculaire, Université Grenoble Alpes, F-38000 Grenoble, France; 4Commissariat à l’Energie Atomique et aux Energies Alternatives CEA, Laboratoire D’électronique et de Technologie de L’information, Département Technologies pour la Biologie et la Santé LETI-DTBS, Université Grenoble Alpes, F-38000 Grenoble, France

**Keywords:** photodynamic therapy, lipid nanoparticles, drug delivery system, tumor vectorization, verteporfin, ovarian carcinomatosis, spheroids

## Abstract

Advanced ovarian cancer is the most lethal gynecological cancer, with a high rate of chemoresistance and relapse. Photodynamic therapy offers new prospects for ovarian cancer treatment, but current photosensitizers lack tumor specificity, resulting in low efficacy and significant side-effects. In the present work, the clinically approved photosensitizer verteporfin was encapsulated within nanostructured lipid carriers (NLC) for targeted photodynamic therapy of ovarian cancer. Cellular uptake and phototoxicity of free verteporfin and NLC-verteporfin were studied in vitro in human ovarian cancer cell lines cultured in 2D and 3D-spheroids, and biodistribution and photodynamic therapy were evaluated in vivo in mice. Both molecules were internalized in ovarian cancer cells and strongly inhibited tumor cells viability when exposed to laser light only. In vivo biodistribution and pharmacokinetic studies evidenced a long circulation time of NLC associated with efficient tumor uptake. Administration of 2 mg·kg^−1^ free verteporfin induced severe phototoxic adverse effects leading to the death of 5 out of 8 mice. In contrast, laser light exposure of tumors after intravenous administration of NLC-verteporfin (8 mg·kg^−1^) significantly inhibited tumor growth without visible toxicity. NLC-verteporfin thus led to efficient verteporfin vectorization to the tumor site and protection from side-effects, providing promising therapeutic prospects for photodynamic therapy of cancer.

## 1. Introduction

Ovarian cancer accounts for about 4% of worldwide cancer incidence and is the most lethal gynecological cancer [1]. Seventy-five percent of ovarian cancer cases are detected in late stages, and the 5-year survival rate of patients in advanced-stages is barely 30% [2]. Due to its non-symptomatic advancement, high metastasis rate, and its resistance to chemotherapy, treatment of ovarian cancer constitutes a clinical challenge. Tumor growth and dissemination of ovarian cancer within the peritoneal cavity result in peritoneal carcinomatosis. Current treatments are based on tumor surgery combined with intravenous chemotherapy [2,3]. Unfortunately, most women with ovarian cancer relapse after the first-line therapy [2]. It is therefore critical to develop new approaches and treatment options to decrease the recurrence rate and to improve patient survival [2,3].

Photodynamic therapy (PDT) is an emerging and promising therapeutic modality for fighting cancer. PDT is based on the administration of a photosensitizer and its activation at the tumor site when exposed to light of a particular wavelength. The activated photosensitizer releases energy to generate highly toxic singlet oxygen (^1^O_2_) and reactive oxygen species (ROS). These products mediate microvascular damages, tumor cell toxicity, and anti-tumor immune responses [4,5,6]. PDT is used for the treatment of several cancers, in particular those that are accessible to laser light including skin, bladder, gastroesophageal, and lung cancers [4]. There are several photosensitizing agents currently approved by the US Food and Drug Administration. Among those, verteporfin (Visudyn^®^) is a second-generation photosensitizer approved for the PDT-based treatment of age related macular degeneration. It is now extensively studied as PDT agent for treatment of numerous cancer types [4,7,8,9]. Verteporfin is activated by near infrared (NIR) light (689 nm) that penetrates deeply into tissues and displays a very high yield of singlet oxygen production [7]. However, this photosensitizer lacks tumor specificity, resulting into adverse effects on healthy tissues and thus limiting its concentration of use [4,7,10]. Combining photosensitizing drugs with nanoparticles could overcome some of the limitations encountered with free photosensitizers [11,12,13], in particular because it is expected to increase their cellular uptake and passive accumulation into tumors due to the enhanced permeability and retention (EPR) effect [12,14]. Verteporfin loaded in mesoporous silica nanoparticles has been shown to inhibit the growth of melanoma cells in vitro [15], and in vivo in subcutaneous tumor bearing mice after topical administration and light exposure [16].

In the present study, verteporfin was encapsulated within nanostructured lipid carriers (NLC) for PDT of ovarian cancer after systemic administration. Lipid-based nanoparticles are used as drug nanocarriers because of their excellent biocompatibility, long circulation time, and high tumor accumulation due to the EPR effect [12,13,17,18,19]. The cell-uptake and phototoxicity of NLC-verteporfin were first assessed in vitro using 2D-monolayers and 3D-spheroids of human ovarian cancer cells. Biodistribution and pharmacokinetic studies of NLC-verteporfin were then evaluated in vivo after intravenous injection. Finally, the PDT efficacy of NLC-verteporfin was compared to free verteporfin in mice with ovarian cancer tumors. This study provides promising therapeutic prospects for PDT in ovarian cancer.

## 2. Results

### 2.1. Nanostructured Lipid Carriers Efficiently Encapsulate Verteporfin

NLC were used to encapsulate the photosensitizer verteporfin, or the NIR fluorophore Lipimage^TM^815 as a drug delivery reporter. NLC formulation was performed as previously described [19], providing neutral nanoparticles with an hydrodynamic diameter of 47.9 ± 1.0 nm and a polydispersity index < 0.2 (Table 1). Verteporfin encapsulation yield in NLC was quantitative (>95%), with a final formulation payload of 943 µg of drug for 100 mg of lipids. This formulation was stable for at least 3 months (size variation < 10%, polydispersity index < 0.2), when stored at 4 °C in dark in water (concentration of ~100 mg·mL^−1^ of lipids) (Table 1).

Considering spectral properties, NLC-verteporfin displayed similar absorption profiles at 690 nm compared to free verteporfin in culture medium, in particular with an absorption peak at 690 nm (Appendix A), and showed that verteporfin molecules were mainly in a monomeric form without aggregation inside the lipid core of the nanoparticles. The loading capability was also confirmed by comparable fluorescence emission signal intensity from verteporfin and NLC-verteporfin after excitation at 420 nm (Appendix A).

### 2.2. Verteporfin and NLC-Verteporfin Bind and are Internalized in Ovarian Cancer Cells

The cellular uptake of verteporfin and NLC-verteporfin was evaluated in three different ovarian cancer cell lines, and was found to be slower for NLC-verteporfin than for free verteporfin. Indeed, flow cytometry analyses showed maximal binding of verteporfin in ovarian cancer cell lines SKOV3, IGROV1, and OVCAR3 after 2 h and 24 h incubation at 37 °C (Figure 1). In contrast, the binding of NLC-verteporfin was increased at 24 h compared to 2 h incubation in these cells. At 4 °C, the internalization process was inhibited for both verteporfin and NLC-verteporfin (Figure 1).

Confocal microscopy showed that both verteporfin and NLC-verteporfin were mostly found in the cytoplasm of SKOV3 and OVCAR3 cells (Figure 2a and Appendix A), confirming their internalization in cells. In SKOV3 spheroids, which mimic solid tumors with appropriate cell–cell interactions as well as gradients of nutrients and oxygen [20], verteporfin, and NLC-verteporfin uptakes were observed in both at the periphery and at the center of the spheroids (Figure 2b).

### 2.3. Verteporfin and NLC-Verteporfin Induce Phototoxicity in Ovarian Cancer Cells and Spheroids

Both free verteporfin and NLC-verteporfin induced high phototoxicity in ovarian cancer cells in vitro. Increasing concentrations of free verteporfin or NLC-verteporfin exposed to NIR light reduced the viability of SKOV3 and OVCAR3 cells cultured in monolayers and in spheroids, whereas they had no toxic effect on their proliferation in the dark (Figure 3). The OVCAR3 cells cultured in monolayer appeared to be less sensitive to verteporfin and NLC-verteporfin treatments as compared to SKOV3 cells (Figure 3a,b and Table 2), but interestingly OVCAR3 spheroids were highly sensitive to light exposure 2 h or 24 h after both treatments, with IC_50_ values similar to those of SKOV3 spheroids (Figure 3c,d, Table 2 and Appendix A).

The drug-light interval (light exposure 2 h or 24 h after treatment) had no influence on the phototoxicity of both treatments in 2D cell cultures (Figure 3a,b and Table 2). In contrast, both verteporfin and NLC-verteporfin had a stronger toxic effect on the viability of SKOV3 and OVCAR3 spheroids when exposed to the light 24 h after treatment as compared to 2 h (Figure 3c,d and Table 2).

### 2.4. NLC Accumulate in Ovarian Tumors In Vivo

The in vivo biodistribution and pharmacokinetics of NLC were assessed with particular interest on tumor uptake and specificity. While free verteporfin was rapidly eliminated from the bloodstream following intravenous administration (elimination half-life: 1.9 h) (Figure 4a), NLC circulated for several hours as shown by ex vivo fluorescence imaging of plasma samples from healthy mice after intravenous injection of fluorescent dye-loaded NLC (NLC-LipImage^TM^815: elimination half-life: 6.5 h) or NLC-verteporfin (elimination half-life: 9.1 h) (Figure 4a). 

In addition, in vivo biodistribution of NLC-LipImage in mice with subcutaneous SKOV3 tumors showed accumulation into the liver, as expected with lipid nanoparticles, reaching a plateau between 5 h and 24 h (Figure 4b,c). Nonetheless, NLC also progressively accumulated into the tumor, reaching a maximum 24 h after injection. Tumor specificity was illustrated by a tumor to skin ratio of 3.1 ± 0.5 at 24 h (Figure 4c). At this time, the mice were euthanized, and the collected organs were subjected to ex vivo fluorescence imaging (Figure 4d), confirming the strong fluorescence signal in the liver, but also the accumulation of NLC-LipImage into the tumor, with a high tumor to muscle ratio of 11.5 ± 0.8 (Figure 4e). High performance liquid chromatography (HPLC), as another quantification method, also confirmed these results in tissues and blood plasma (Figure 4f and Appendix A). Altogether, in vivo and ex vivo fluorescence imaging demonstrated long circulation time, liver uptake, and tumor accumulation of NLC-verteporfin after intravenous injection in mice.

### 2.5. NLC-Verteporfin Improves PDT, Free Verteporfin is Highly Toxic In Vivo

Mice with well-established subcutaneous SKOV3 ovarian tumors were treated by single intravenous injection of free verteporfin, NLC-verteporfin or vehicle (control group) (Figure 5a). Tumors were then exposed to NIR laser light and the tumor volume was monitored for 2 weeks.

We first assessed the PDT efficacy of free verteporfin. When tumors were exposed to laser light 15 min after free verteporfin injection (2 mg·kg^−1^), their growth was rapidly and strongly reduced (Figure 5b). However, three mice died on the day of light exposure, and two others died one week after, showing the strong toxicity of circulating activated verteporfin (Figure 5b and Appendix A). Among the three remaining mice at the end of the experiment, two were totally cured, and one escaped the treatment. In contrast, in absence of light or when tumors were exposed to laser light 24 h after verteporfin injection, no effect was observed on either tumor growth or animal wellbeing, showing the high photospecificity and the short circulation half-life of verteporfin, respectively (Figure 5b and Appendix A). Similarly, intravenous administration of NLC-verteporfin (2 mg·kg^−1^ of verteporfin), in the dark or when tumors were exposed to laser light 24 h after NLC-verteporfin administration, was safe and did not induce any sign of toxicity. However, no effect on tumor growth was observed, suggesting inadequate PDT conditions (Figure 5c).

Since no undesirable toxicity had been observed using NLC-verteporfin (Appendix A), a new experiment was performed using higher NLC-verteporfin concentration and higher light fluency. We checked the innocuousness of light exposure at 200 J·cm^−2^ in a preliminary experiment, and observed no effect on mice tumor and skin (Appendix A). Thereby, NLC-verteporfin (8 mg·kg^−1^ of verteporfin) was combined with 200 J·cm^−2^ light exposure 24 h after intravenous administration, and significantly inhibited tumor growth as compared to the control group or to NLC-verteporfin in the dark (Figure 5d). These treatments were well tolerated and no significant weight loss was observed (Appendix A). In accordance with our preliminary experiment, the 200 J·cm^−2^ light exposure did not damage the skin (Appendix A). Moreover, in most tumors treated with NLC-verteporfin (7/8 tumors), a whitening of the tumor was observed during the days following light exposure that might indicate treatment activity (Appendix A). In addition, necrotic areas were observed in tumors treated with NLC-verteporfin combined with laser light only (Figure 5e). Macroscopic observation of the main organs did not show evidence of damage. In addition, liver sections did not show any evidence of necrosis or damage (Appendix A).

Besides, tumor tissue oxygen saturation (StO_2_) measured by non-invasive photoacoustic imaging 24 h after light exposure was decreased among mice treated with NLC-verteporfin as compared to the control group (Appendix A), suggesting vascular impairment resulting in tissue hypoxia in the treated tumors. Interestingly, the two tumors from the treated group displaying the higher StO_2_ values were the less-responsive tumors (Appendix A). 

Taken together, these results showed that NLC-verteporfin accumulation in tumor allowed a significant inhibition of tumor growth after light exposure with high fluency, and decreased side-effects.

### 2.6. NLC Accumulate in Disseminated Ovarian Tumor Nodules

Peritoneal carcinomatosis results from tumor growth and dissemination within the peritoneal cavity, and disseminated tumor nodules could be treated with PDT [3]. To evaluate tumor accumulation and drug delivery in disseminated small tumor nodules, NLC tumor uptake was assessed in an orthotopic mouse model of human peritoneal carcinomatosis from ovarian cancer. Luciferase-modified SKOV3 ovarian cancer cells were inoculated into one ovary, and tumor growth and peritoneal dissemination was monitored using non-invasive bioluminescence imaging in vivo (Figure 6a). When peritoneal carcinomatosis was established (4 to 5 weeks after tumor cells implantation), NLC-LipImage was administered intravenously. Despite the predictable high liver signal, NLC-LipImage provided clearly detectable signal spots in the peritoneal cavity (Figure 6b, Video S1 and Video S2). Furthermore, intraoperative fluorescence imaging was used simultaneously with bioluminescence imaging, and showed the colocalization of NLC with most tumor nodules (Figure 6c), demonstrating NLC tumor uptake in peritoneal carcinomatosis.

## 3. Discussion

In this study, we showed that the encapsulation of the clinically approved photosensitizer verteporfin in NLC allowed cellular uptake and high phototoxicity in ovarian cancer cells cultured in monolayers and in 3D-spheroids. When injected intravenously in mice, these nanoparticles demonstrated tumor uptake, including in peritoneal small tumor nodules. Furthermore, NLC-verteporfin improved PDT efficacy in ovarian tumors in vivo with no phototoxic adverse effects compared to free verteporfin.

Lipid-based nanoparticles appear as nanocarriers of choice for lipophilic molecules such as the verteporfin photosensitizer [13,19]. We demonstrated here that verteporfin loading did not affect the colloidal properties of NLC. In addition, NLC showed high verteporfin encapsulation efficiency and stability for several months, as previously shown [21]. With solvent-free simple up-scalable fabrication process, NLC-verteporfin could easily be produced in large quantities for the clinic.

The cellular uptake of NLC-verteporfin involved endocytosis processes, taking more time than free verteporfin, part of which can rapidly diffuse across cell membranes. NLC-verteporfin uptake was also shown in tumor spheroids, which mimic in vivo tumors with tumor cell interactions and gradients of nutrients and oxygen [22,23,24,25]. The slower kinetic of NLC-verteporfin uptake compared to free verteporfin did not impair the efficiency of verteporfin phototoxicity. Indeed, verteporfin had the same phototoxicity when free or loaded in NLC in both SKOV3 and OVCAR3 ovarian cancer cells and spheroids. Accordingly, the phototoxic activity of verteporfin formulated in nanoparticles has recently been shown in ovarian carcinoma cells in vitro [26,27]. In addition, we observed that PDT was more efficient in spheroids than in monolayer-cultured cells, especially on OVCAR3 spheroids. The mechanism by which the OVCAR3 cells resisted to PDT in 2D culture but not in spheroids is still unknown. These results highlighted that the toxicity of drugs significantly varied depending on whether it is assessed in spheroids or in monolayer cell cultures, and this thus underlined the importance of using spheroids before moving forward in vivo experiments in mice [22,23,25]. Further experiments are needed to study NLC-verteporfin uptake and cytotoxicity in multicellular spheroids, which will reproduce both tumor cells and microenvironment [24]. Taking into account cellular and microenvironment heterogeneity within tumors is particularly important to enhance the accumulation of nanoparticles in the tumor.

Verteporfin therapy is the first-line therapy for serious ocular diseases, age-related macular degeneration and myopic choroidal neovascularization [28]. In these cases, the illumination of the retina is performed 15 min after intravenous injection of the photosensitizer, and the photodynamic reaction produces an anti-vascular effect that reduces disease progression. Because of its high NIR light-specific phototoxicity (in the nanomolar range), verteporfin is also extensively studied as PDT agent for the treatment of cancers, but mainly in vitro at the moment [4,7,9,27]. Yet, verteporfin has been evaluated in a phase I/II clinical study in patients with advanced pancreatic cancer, and has been shown to induce PDT-dependent tumor necrosis with high variability [8]. In our work, illumination of subcutaneous SKOV3 ovarian tumors 15 min after intravenous injection of a high dose of verteporfin led to strong tumor regression, illustrating the strong phototoxic effect of verteporfin on vasculature, but was associated with dramatic adverse effects and ultimately with death of most of the mice. The inhibitory effect of verteporfin on tumor vasculature and/or tumor cells, and its relationship to side-effects, remain to be formally investigated. Given the high photospecificity of verteporfin, no effect was observed in the absence of light. Furthermore, according to the short circulation half-life of free verteporfin, we did not observe any impact on tumors exposed to laser light 24 h after verteporfin injection. Altogether, this strongly suggested the need to formulate verteporfin to improve its tumor specificity and to decrease its side-effects.

A major issue for PDT is therefore to improve photosensitizer delivery efficiency and tumor specificity [7]. Many nanoparticle-based formulations have been proposed and developed, in order to exploit the EPR effect of tumors [7,13,14]. According to our data, and to the literature [12,13,17,18,19], NLC had a long circulation time and accumulate in subcutaneous tumors, as well as in peritoneal small tumor nodules 24 h after injection. This accumulation enabled tumor delivery of high concentrations of verteporfin, as demonstrated by tumor fluorescence imaging and HPLC dosing. These data and previous study [29] suggested that NLC were rapidly dissociated following their internalization in tumor cells, thus releasing verteporfin. Furthermore, photoactivation was performed 24 h after injection when circulating NLC-verteporfin concentration was low. This appeared to protect from post-illumination systemic adverse effects of verteporfin, since no toxicity was observed. In addition, no evidence of toxicity was observed in liver, despite the high hepatic accumulation of NLC, as it has been previously shown with dye-loaded NLC [30]. Nanoparticles-based encapsulation of verteporfin has been shown to promote anticancer activity in melanoma cells in vitro [15] and in vivo after topical administration and repeated light exposure [16]. In addition, co-encapsulation of verteporfin with chemotherapies, such as temozolomide in glioblastoma multiform cells in vitro [31], cisplatin in SKOV3 cells in vitro [32], or docetaxel in colon cancer cells in vitro and in vivo after six instances of light exposure [33], has also shown anticancer activity. Our study is the first, to our knowledge, to show PDT efficacy in subcutaneous ovarian tumors after single intravenous administration of NLC-verteporfin and laser light exposure. These results should be reinforced by using orthotopic model of peritoneal carcinomatosis from ovarian cancer.

The success of PDT mainly depends on total light dose, exposure time, and fluency rate [34]. Here, we demonstrated that tumor growth inhibition was related to the combination of NLC-verteporfin dose (8 mg·kg^−1^ of verteporfin) and laser light fluency (690 nm and 200 J·cm^−2^), these factors yet needing further optimization. Optimal excitation wavelengths for PDT are described between 600 and 850 nm, and a fluency of 200 J·cm^−2^ is usually used to avoid thermal effect [34]. Our data showed that these PDT conditions induced tumor necrosis. A specific quantification of tumor necrosis should be performed to confirm these observations. However, in our experiment, although PDT significantly inhibited tumor growth, tumors still grew. A white coloring was observed on most of the tumors following light exposure, but it disappeared after 5 days, suggesting that some tumor cells escaped PDT or were incompletely treated, especially at the tumor periphery. With our illumination set-up, the area optimally covered by the laser light was 5 mm in diameter, and may not be large enough to treat the entire tumor (6.4 ± 1.0 mm mean diameter at the day of illumination). Tumor periphery thus received suboptimal illumination, which can explain the partial response and consequent relapse we observed. Delivering the activating light to allow PDT to uniformly treat all tumor cells is indeed challenging, but essential for optimal therapeutic efficacy [3,35,36]. In addition, NLC-verteporfin efficacy may be transient. Interestingly, repeating PDT, as experienced in colon cancer or melanoma [16,33], might reduce the risk of tumor recurrence with no accumulative toxicity, hypersensitivity or resistance [3,36]. This yet requires repeated access to the tumor, which may not be practical, depending on the tumor location and accessibility. Altogether, these data suggested that the dose regimen of NLC-verteporfin have to be optimized to achieve long-term antitumor efficacy. Furthermore, the effect of PDT on induction of inflammatory and immune response, as well as its combination with cancer immunotherapies might be investigated to enhance the PDT response [5,6,13,34].

We further used non-invasive photoacoustic imaging to measure tissue oxygenation in the tumor 24 h after PDT treatment. The reduced levels of StO_2_ observed in tumors responding to the treatment suggested vascular impairment and tissue hypoxia, although this remains to be demonstrated by CD31 staining in tumors. In contrast, high levels of StO_2_ were observed in tumors not responding to treatment. This suggested that tumor hypoxia monitoring by non-invasive photoacoustic imaging could be used to predict early tumor response to PDT, as soon as 24 h after light exposure. Similarly, photoacoustic imaging has been previously demonstrated to be predictive of tumor response to radiotherapy in head and neck cancer [37].

Our results offer new prospects for the management of ovarian cancer and in particular for the treatment of peritoneal carcinomatosis, in which conventional cytoreductive surgery often remains suboptimal for a number of patients [2,3]. Indeed, some tumor residues cannot be surgically removed because of their location or close contact with vital structures, and are responsible for relapse. Actually, PDT has already been demonstrated to be combined with conventional surgical tumor resection to improve treatment outcome of peritoneal carcinomatosis despite a narrow therapeutic index [3]. Intraoperative PDT could be applied immediately following surgical tumor debulking and could treat residual peritoneal tumors in areas where surgical procedures pose a high risk of perioperative complications [38,39]. Significant toxicity induced by intraperitoneal PDT has been shown in Phase I and II clinical trials, owing to the heterogeneous population of patients, who had poor prognosis and failed to respond to first-line treatments, and to the non-specificity of photosensitizers for tumor cells [3]. We showed that NLC-verteporfin accumulated in small tumor nodules and provided tumor contrast that can be detected by intraoperative fluorescence imaging. Therefore, photoactivating laser light can be specifically delivered on identified tumor residue, thus further increasing treatment specificity and sparing healthy tissue, including the liver. Treating residual peritoneal metastases by PDT after cytoreductive surgery should thus be further investigated depending on metastatic tumor burden. 

## 4. Materials and Methods

### 4.1. Materials

Suppocire™ NB was purchased from Gattefossé (Saint-Priest, France). Myrj^TM^ S40 (poly(ethylene glycol) stearate surfactant with 40 ethylene glycol motifs, and Super-refined Soybean oil were supplied by CRODA (Chocques, France). Lipoid^TM^ s75 was purchased from Lipoid GmbH (Ludwigshafen, Germany). HPLC grade solvents were obtained from VWR Scientific (Fontenay sous Bois, France) and other chemicals including verteporfin from Sigma-Aldrich (Saint-Quentin Fallavier, France). 

HPLC-grade water (specific resistance = 18.2 MΏ cm) was obtained from a Classic DI MK2 water purification system (Elga, UK) and was used in all experiments. SpectraPor™ dialysis membrane 12–14,000 was purchased from Roth Sochiel EURL (Lauterbourg, France). Nanostructured lipid carriers (NLC) were formulated using a VCX750 Ultrasonic processor from Sonics (Newtown, CT, USA) equipped with a 3 mm diameter microtip.

### 4.2. Formulation of Nanostructured Lipid Carriers

The lipid phase was prepared by mixing solid (Suppocire™ NB, 255 mg) and liquid (Super-refined Soybean oil, 85 mg) lipids as well as the lipophilic surfactant Lipoid™ s75 (65 mg) and verteporfin (4 mg) while the aqueous phase was composed of the hydrophilic surfactant, Myrj^TM^ S40 (345 mg) prepared in 1250 µL DI water. After homogenization at 50 °C, both lipid and aqueous phases were crudely mixed and sonication cycles were performed at 45 °C during 5 min. Non-encapsulated components were separated from NLC by dialysis (1× PBS, MWCO: 12–14 kDa, overnight). Dye-loaded NLC (LipImage™ 815) were synthetized by a similar process, as previously described [40]. Prior to characterization and injection, NLC dispersions were filtered through a 0.22 µm cellulose Millipore membrane for sterilization. Particle concentration was assessed by weighing freeze-dried samples of NLC obtained from a known volume. NLC formulation was stored in liquid suspension at 4 °C in dark.

### 4.3. Dynamic Light Scattering 

The hydrodynamic diameter and zeta potential of the NLC were measured at 22 °C with a Malvern Zeta Sizer Nanoinstrument (NanoZS, Malvern, UK) in 0.1× PBS buffer. Physical stability was investigated by DLS measurements over 3 months with samples stored at 4 °C in dark. Mean average diameters and polydispersity indices reported were obtained from scattered light intensity results. Data were expressed in terms of mean and standard deviation of three independent measurements.

### 4.4. HPLC Analyses

Verteporfin was analyzed by HPLC using a Waters Alliance 2695 Separation module equipped with a Waters 2487 dual absorbance detector and a reverse-phase C18 column (Atlantis T3, 5 µm, 50 × 4.6 mm, 100 Å) (Waters, Milford, MA, USA). Data were processed using Empower™ 2 software. The elution used a gradient mobile phase consisting of (A) 0.05 mM monobasic sodium phosphate (pH 3.5) solution and (B) methanol, according to previously published method [41]. The gradient was applied as follows: 0–4 min from 50 to 65% eluent B; 4–8 min from 65 to 85% eluent B; 8–22 min from 85 to 99% eluent B; 22–27 min from 99 to 10% eluent B; 27–30 min from 10 to 50% eluent B; 30–40 min 50% eluent B. A flow rate of 1 mL/min was used with a 20 µL injection volume. Verteporfin was detected at 430 nm (Soret band) and linear calibration curve was established from 0 to 600 µg.mL^−1^ for quantification.

For analysis of the verteporfin payload in NLC, nanoparticles were disassembled. 100 µL of NLC was added to 100 µL of methanol. Samples were centrifuged to pellet lipids after precipitation, and the supernatant was taken for analysis. The lipid pellet was rinsed three times more with 100 µL methanol. The total supernatant, adjusted to 500 µL, was centrifuged once more to remove any remaining lipids and injected to HPLC analysis for verteporfin quantitation.

For plasma pharmacokinetics, plasma samples were diluted two-fold with (MeOH 90%/NaH_2_PO_4_ 0.05 mM 10%) and centrifuged for 1h at 13,000 rpm. The supernatant was injected on HPLC. For organ quantification, tumors, livers and kidneys were crudely crushed then incubated with 1 mL of (MeOH 90%/NaH_2_PO_4_ 0.05 mM 10%), for 4 days. After centrifugation for 1 h at 13,000 rpm, the supernatant was analyzed by HPLC.

### 4.5. Absorbance and Fluorescence Spectrum

Evolution 201 UV–visible spectrophotometer (Thermo Fisher Scientific, Waltham, MA, USA) and LS 55 Fluorescence Spectrometer (PerkinElmer, Waltham, MA, USA) were used to measure verteporfin and NLC-verteporfin absorbance in medium and fluorescence in 1× PBS, respectively. 

### 4.6. Cell Lines and Culture

The human SKOV3 and OVCAR3 cell lines were obtained from LGC standard (Molsheim, France). The human IGROV1 cell line was obtained from Institut Gustave Roussy (Villejuif, France). All cells were routinely tested for the presence of mycoplasma (Mycoalert^®^ Mycoplasma Detection Kit, Lonza, Rockland, ME, USA) and used within three months after thawing.

#### 4.6.1. Two-Dimensions (2D) Cell Culture

Cells were maintained in culture at 37 °C in RPMI-1640 medium with 10% FBS (SKOV3 and IGROV1) or with 20% FBS and 0.01 mg·mL^−1^ insulin (OVCAR3) in a 5% CO_2_ humidified atmosphere. The cell morphology was routinely checked.

#### 4.6.2. Three-Dimensions (3D) Cell Culture

Spheroids were generated by plating 3000 cells/well, into 96-well round bottom ultra-low attachment (ULA) spheroid microplates (Corning, Tewksbury, MA, USA). The spheroid culture was performed in complete medium in a humidified atmosphere with 5% CO_2_. Spheroid formation and growth were assessed by microscopic examination using an inverted microscope and by imaging the spheroids at each time point. 

### 4.7. Flow Cytometry

Cells were plated on 6-well plates and treated with verteporfin or NLC-verteporfin for 2 h or 24 h at 37 °C or at 4 °C in the dark. After incubation cells were harvested and re-suspended in 1× PBS for flow cytometry analysis. Fluorescence emission of verteporfin was analyzed using flow cytometry LSRII and FCS Express software (BD Biosciences, San Jose, CA, USA).

### 4.8. Fluorescence Microscopy

Cells were plated on labtek^®^ and treated with verteporfin or NLC-verteporfin for 2 h or 24 h in the dark at 37 °C before confocal imaging. Spheroids were treated with verteporfin or NLC-verteporfin incubation for 24 h at 37 °C, and harvested, washed, fixed in 4% paraformaldehyde, frozen in optimal cutting temperature (OCT) compound for embedding, and cut into 12-μm sections. Hoechst was used to counterstain the cell nuclei. Fluorescence microscopy was carried out using a confocal microscope (LSM 710; Carl Zeiss, Jena, Germany). An objective Plan Apochromat 20×/0.8 NA in air and an objective Plan Apochromat 63×/1.4 NA in oil were used. The excitation was at 405 nm and emission filter between 650 and 730 nm for verteporfin, and between 450 and 500 nm for Hoechst.

### 4.9. In Vitro Cytotoxicity Assays

Cell proliferation assays were conducted in 96-well culture plates. 2D-cultured cells and spheroids were cultured for 24 h and 72 h, respectively, prior to treatment with increasing concentration of verteporfin or NLC-verteporfin in complete medium. Illumination at 690 nm for 60 s with power output at 167 mW·cm^−2^ (fluency 10 J·cm^−2^, High Power Devices Inc., North Brunswick, NJ, USA) was performed 2 h or 24 h after treatment. The cell viability was quantified 72 h after light exposure using Presto Blue^TM^ (Invitrogen, Waltham, MA, USA) in 2D cell models, or using the CellTiter-Glo^®^ 3D Cell Viability Assay (Promega, Charbonnière, France) in spheroids. Fluorescence was measured using a multilabel counter (Omega Fluostar, BMG Labtech, Champigny sur Marne, France) with the excitation and emission filter at 560 and 590 nm, and the bioluminescence was measured using counter (Victor^3TM^, Perkin Elmer, Waltham, MA, USA). Cell viability was normalized to control cells (no drug and un-illuminated). The drug concentrations required to inhibit cell growth by 50% (IC_50_) were interpolated from the dose-response curves.

### 4.10. In Vivo Experiments

All animal studies were performed in accordance with the European Economic Community guidelines and the “Principles of Laboratory Animal Care” (NIH publication N 86-23 revised 1985) and were approved by the institutional guidelines and the European Community (EU Directive 2010/63/EU) for the use of experimental animals (authorization for the experiment: apafis#15176-201808281433752 v1).

#### 4.10.1. Subcutaneous Ovarian Tumor Model

Anesthetized (4% isoflurane/air for anesthesia induction and 1.5% thereafter) six-week-old female NMRI nude mice (Janvier Labs, Le Genest-Saint Isle, France) were injected subcutaneously in the flank with 10^7^ luciferase-modified SKOV3 cells in 1× PBS. Tumor size was measured three times a week using a caliper, and the tumor volume was calculated as
length × (width)^2^ × 0.4

#### 4.10.2. Pharmacokinetics Studies on Blood Plasma Samples

Healthy mice or mice with SKOV3 subcutaneous tumor were anesthetized using 4% isoflurane/air for anesthesia induction and 1.5% thereafter, and 200 µL NLC-LipImage^TM^815 (50 µM of Lipimage^TM^815) or NLC-verteporfin (8 mg·kg^−1^ of verteporfin) or free verteporfin (2 mg·kg^−1^) were administered intravenously via the tail vein. Fifty microliters of blood were sampled from the tail vein before and at different times after injection, centrifuged 5 min at 8,000 g, and 10 µL of plasma were used for fluorescence imaging [23,42] or HPLC. Half-lives were measured from nonlinear regression fit analyses (two phase decay). Tumor, liver, and kidney samples were excised 24 h after treatments, and frozen for further HPLC analyses.

#### 4.10.3. Biodistribution of Fluorescent NLC In Vivo

2D-fluorescence images were acquired at several time points after intravenous administration of NLC-LipImage^TM^-815 (200 µL, 50 µmol·L^−1^) using the Fluobeam800^TM^ (Fluoptics, Grenoble, France) that excites fluorescence at 780 nm and detects emitted light at wavelengths greater than 830 nm. At the end of the experiment, mice were sacrificed, and some organs were collected for ex vivo imaging using the Fluobeam800^TM^. Semiquantitative data were obtained using the Wasabi^®^ software (Hamamatsu, Massy, France) by drawing regions of interest (ROIs) on the different organs and were expressed as the number of relative light units per pixel per unit of exposure time and relative to the fluorescence signal in the skin or muscle [22,42].

#### 4.10.4. Orthotopic Murine Model of Peritoneal Carcinomatosis from Ovarian Cancer

Six-week-old female NMRI nude mice (Janvier) were anesthetized (isoflurane/air 4% for induction and 2% then after) and placed on a heating mat at 37 °C. Buprenorphin 0.6 mg·kg^−1^ was injected subcutaneously for analgesia before and 16 h after surgery. An incision of 2 cm was performed in the peritoneum. Luciferase-modified SKOV3 cells were suspended in medium at 3 × 10^6^ cells/50 µL and were slowly injected into the ovaries through lumen of the fallopian tube. The peritoneum and the skin were closed with synthetic absorbable suture (Optime^®^ 4.0 for the peritoneum and Monocryl^®^ 4.0 for the skin), and mice were placed on the right flank and were carefully observed until they woke up. Tumor growth and peritoneal invasion were weekly monitored by in vivo bioluminescence imaging (IVIS KINETIC, Perkin Elmer, Waltham, MA, USA) 5 min after the intraperitoneal injection of 150 mg·kg^−1^ of D-luciferin (Promega, Charbonnière, France).

Fluorescence diffuse optical tomography (fDOT) acquisitions were performed 24 h after intravenous administration of NLC-LipImage^TM^-815 (200 µL, 50 µM) in mice with established peritoneal carcinomatosis or in healthy mice. This system consists of a 690 nm laser source, a CCD camera and a set of filters [42]. The light source is a 35-mW compact laser diode (Power Technology, Little Rock, AR, USA) equipped with a bandpass interference filter (685AF30OD6; Melles Griot, Albuquerque, NM, USA). Emitted fluorescence was filtered by 730/30 nm band-pass filter (RG9 OD5; Schott, Mainz, Germany) placed in front of a near infrared-sensitive CCD camera (Hamamatsu Photonics K.K., Shizuoka, Japan) mounted with a f/15-mm objective (Schneider Kreutznach, Bad Kreuznach, Germany). X-ray micro-computed tomography was performed using the vivaCT40 (Scanco Medical, Wangen-Brüttisellen, Switzerland) at 45 kV voltage and 177 µA intensity with an 80 μm isotropic voxel size and 200 ms integration time. 3D fluorescence and microCT reconstructed volumes were merged using external as well as anatomical landmarks from both imaging modalities. Immediately after 3D non-invasive fluorescence imaging, mice received an intraperitoneal injection of 150 mg·kg^−1^ of D-luciferin and were then quickly dissected so as to expose the main organs for sequential bioluminescence and fluorescence imaging (IVIS KINETIC (Perkin Elmer, Waltham, MA, USA) and Fluobeam^®^800 (Fluoptics, Grenoble, France)). 

#### 4.10.5. PDT In Vivo

When SKOV3 subcutaneous tumors of approximately 100 mm^3^ were established (17 days after cells implantation), mice were distributed into groups of 8. Intravenous administration of 200 µL of free verteporfin (1.8 mg·kg^−1^) or loaded in lipids nanoparticles (NLC-verteporfin, 2 or 8 mg·kg^−1^ of verteporfin) was performed in anesthetized mice (isoflurane/air 4% for induction and 1.5% thereafter). Control mice received the vehicle (1× PBS).

Either 15 m or 24 h after verteporfin or NLC-verteporfin injections, subcutaneous tumors were exposed to a near infrared laser emitting at 690 nm with a collimator (convex lens, 25 mm diameter) positioned at 10 cm distance from the tumor, for 5 min and 30 s (150 mW·cm^−2^, fluency 50 J·cm^−2^) or for 22 min and 15 s (150 mW·cm^−2^, fluency 200 J·cm^−2^). Injected groups and control group were then maintained in the dark for 6 days after injection. Mice were observed daily and weighed three times a week. Tumor size was measured three times a week using a caliper for 2 weeks after PDT. Mice bearing tumors ≥ 1.5 cm^3^ were euthanized immediately. At the end of the experiment, tumors were excised and frozen for further analyses.

#### 4.10.6. Photoacoustic Imaging

Photoacoustic imaging was performed 24 h after tumor exposure to light with the Vevo^®^LAZR system (Fujifilm, Visualsonics Inc., Toronto, ON, Canada) using the LZ201 transducer (256 elements linear array; 15 MHz center frequency, [9–18 MHz bandwith], 100 μm axial resolution; 220 μm lateral resolution; 30 × 30 mm^2^ image size). All imaging experiments were conducted under gaseous anesthesia (isoflurane/air 4% for induction and 1.5% thereafter). 3D B-mode and oxyhemo (750 and 850 nm) scans were performed on the subcutaneous tumors. Total oxygen saturation rate (StO_2_) was calculated from oxyhemo data as previously described [43].

#### 4.10.7. Histology

Tumors and livers were frozen and sections of a 7-μm thickness were stained with hematoxylin and eosin (HE). Sections were observed under a Zeiss AxioImager M2 microscope by two researchers who were blinded to the treatment groups (four different samples per group).

### 4.11. Statistical Analyses

All analyses were performed using the GraphPad Prism software (GraphPad Software Inc., San Diego, CA, USA). Statistical comparisons between two groups or more were conducted with Mann–Whitney test, Kruskal–Wallis test, or Friedman test with Dunn’s multiple comparisons post hoc test. Statistical comparisons between mice groups over time were determined by two-way ANOVA with Tukey’s post hoc test. Statistical significance was defined for *p* values ≤ 0.05.

## 5. Conclusions

PDT is increasingly recognized as an emerging clinical tool in cancer therapy besides other therapies. There are several advantages for PDT, such as spatiotemporal control of treatment induction by light activation, reduced systemic cytotoxicity, conventional therapies synergy, chemoresistance reversing, and activation of the immune response. Our study established that the encapsulation of the photosensitizer verteporfin in NLC enhanced PDT in ovarian cancer cells cultured in monolayer, in spheroids, and in vivo in tumor bearing mice. Encapsulating verteporfin in NLC led to higher tumor specificity with both higher verteporfin concentration delivered to the tumor and lower circulating verteporfin at the time of illumination, thus protecting from photosensitizer systemic adverse effects. These results provided promising therapeutic perspectives for PDT in ovarian cancers and in peritoneal carcinomatosis from ovarian cancer in combination with conventional surgery. 

## Figures and Tables

**Figure 1 cancers-11-01760-f001:**
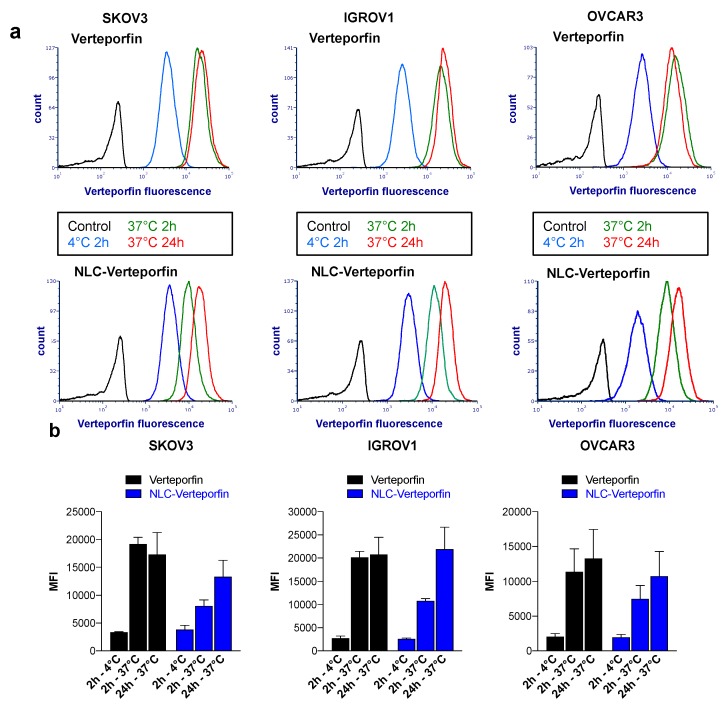
Verteporfin and NLC-verteporfin interact with ovarian cancer cells. Ovarian cancer SKOV3, IGROV1, and OVCAR3 cells were incubated with 1 µmol·L^−1^ verteporfin or NLC-verteporfin for 2 h or 24 h at 4 °C or 37 °C as indicated. (**a**) Histograms show cellular uptake assessed by flow cytometry. Black, control cells; blue, 4 °C for 2 h; green, 37 °C for 2 h; red, 37 °C for 24 h. (**b**) Median fluorescence intensity (MFI). Data are expressed as the mean ± standard deviation (SD) of three independent experiments.

**Figure 2 cancers-11-01760-f002:**
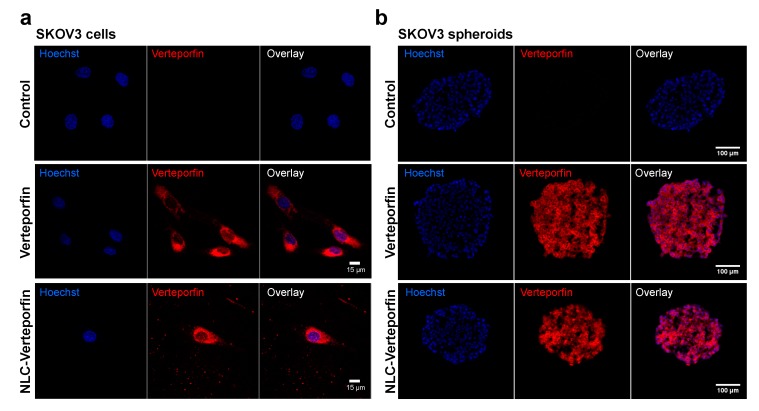
Verteporfin and NLC-verteporfin are internalized in SKOV3 cells and spheroids. Representative confocal microscopy images of SKOV3 cells (**a**) or sections of SKOV3 spheroids (**b**), incubated with 1 µmol·L^−1^ verteporfin or NLC-verteporfin for 24 h. Nuclei are stained with Hoechst 33342 (in blue). Verteporfin fluorescence is observed in red. Control: untreated cells or spheroids.

**Figure 3 cancers-11-01760-f003:**
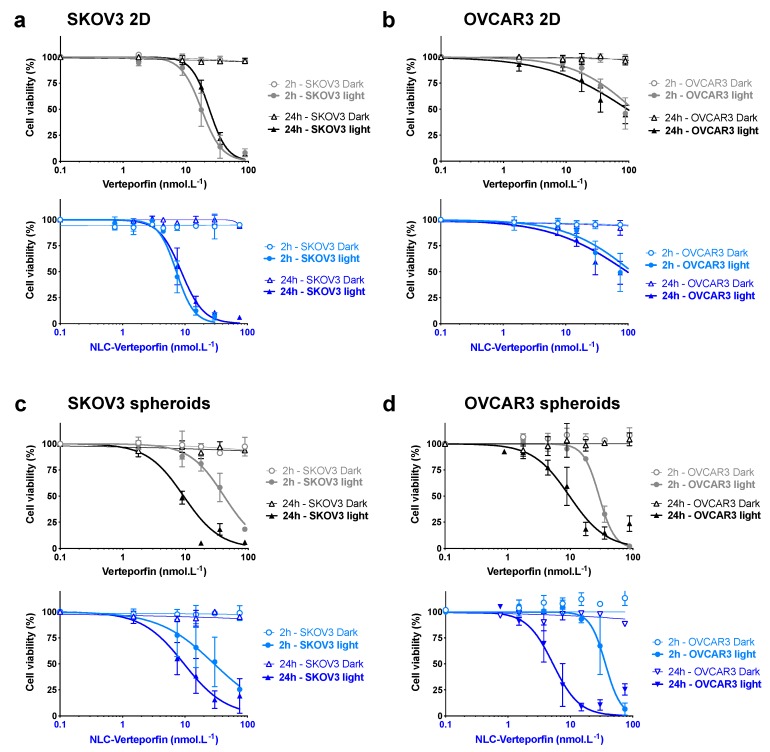
Verteporfin and NLC-verteporfin mediated phototoxicity in SKOV3 and OVCAR3 cells and spheroids. SKOV3 (left panels) and OVCAR3 (right panels) cells (**a**,**b**) or spheroids (**c**,**d**) were treated with increasing concentrations of free verteporfin (in grey/black) or NLC-verteporfin (in blue) for 2 h or 24 h before NIR light exposure at 690 nm (fluency 10 J·cm^−2^). In parallel, cells or spheroids were maintained in the dark. (**a**,**b**) Cell viability was assessed 72 h following light exposure. Data are expressed as the mean ± SD of ≥ 3 independent experiments. (**c**,**d**) Cell viability in spheroids was assessed 72 h following light exposure. Data are expressed as the mean ± SD of ≥ 3 independent experiments in triplicate.

**Figure 4 cancers-11-01760-f004:**
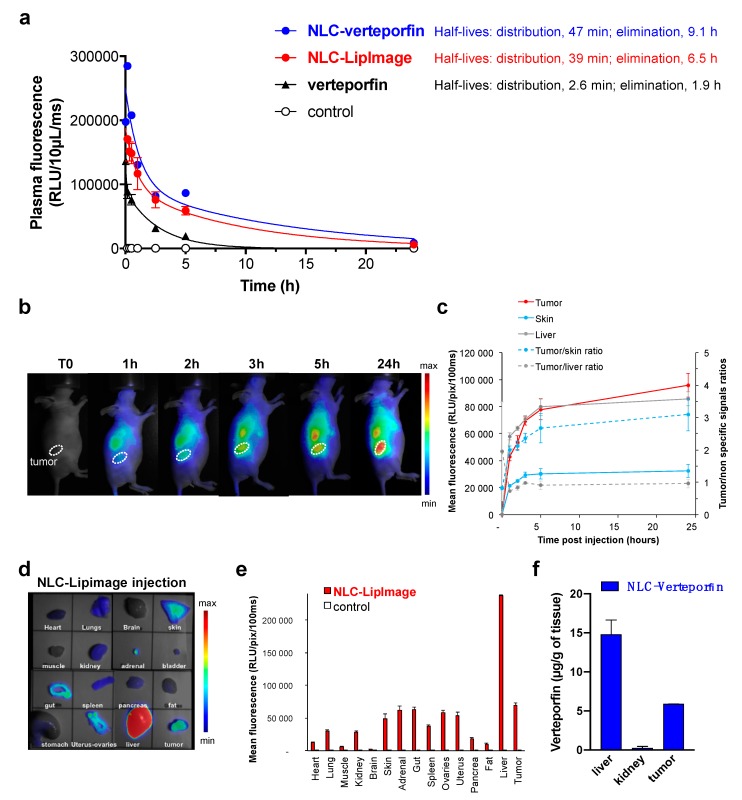
NLC circulated in the bloodstream and accumulated in subcutaneous SKOV3 tumors. (**a**) Healthy mice were injected intravenously with NLC-verteporfin (8 mg·kg^−1^ of verteporfin), dye-loaded NLC (LipImage^TM^-815), or free verteporfin (2 mg·kg^−1^). Fluorescence intensity measurements of verteporfin (for free verteporfin or NLC-verteporfin), or of LipImage^TM^-815 (for NLC-LipImage) were performed on blood plasma samples taken at different time points. The results are expressed as the mean ± SD (*n* = 3). (**b**–**e**) Mice with SKOV3 subcutaneous tumors were injected intravenously with NLC-LipImage. (**b**) Representative fluorescence images (50 ms integration time) recorded using 2D-Fluorescence reflectance optical imaging (FRI) at different times after intravenous injection are shown (min-max: 4243-55869). Dotted lines show subcutaneous tumors. (**c**) Regions of interest (ROI) are defined on tumor, liver, and skin to semi-quantify the amount of photons detected per pixel. The results are expressed as the mean fluorescence ± SD in tumor, skin, and liver, and as the mean tumor/skin and tumor/liver fluorescence ratios ± SD (*n* = 3). (**d**) Fluorescence images were performed on isolated organs 24 h after intravenous injection of NLC-LipImage. Representative fluorescence images of organs in injected and non-injected mice are shown (20 ms integration time; min-max: 1414-27109). (**e**) ROI are defined on the extracted organs to semi-quantify the amount of photons detected per pixel. The results are expressed as the mean ± SD in SKOV3 tumor-bearing mice (*n* = 3) and non-injected mice (control). (**f**) Mice with SKOV3 subcutaneous tumors were injected intravenously with NLC-verteporfin (8 mg·kg^−1^ of verteporfin). Verteporfin measurements were performed by HPLC at 24 h on tumors, livers, and kidneys. The results are expressed as the mean ± SD (*n* = 2).

**Figure 5 cancers-11-01760-f005:**
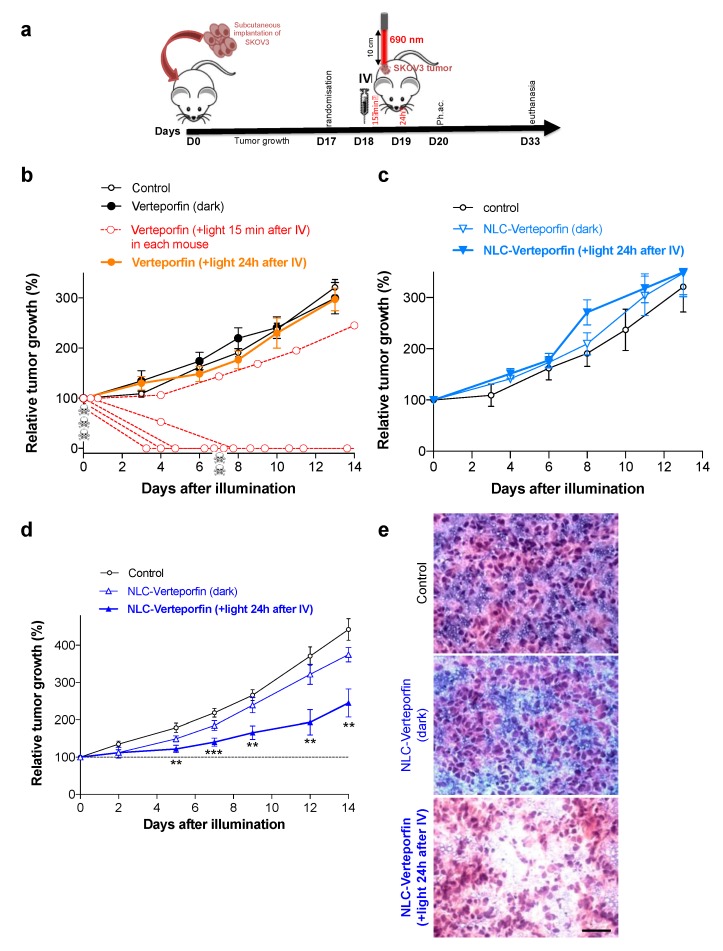
NLC-verteporfin improved PDT after intravenous administration in mice with ovarian cancer. (**a**) Mice with well-established SKOV3 subcutaneous tumors (125 ± 7 mm^3^) were distributed into groups of 8 mice and treated by a single intravenous injection of free verteporfin, NLC-verteporfin or vehicle (control). Mice were maintained in the dark or submitted to a single NIR laser exposure 15 min or 24 h after injection. Non-invasive monitoring of tissue oxygenation (StO_2_) was performed by in vivo photoacoustic imaging (Ph.ac.) on the tumors 24 h after light exposure. The tumor volume was evaluated by caliper measurements 3 times a week for 2 weeks. (**b**,**c**) Mice were injected with 1× PBS (control), free verteporfin (2 mg·kg^−1^), or NLC-verteporfin (2 mg·kg^−1^ of verteporfin), and tumors were exposed to 50 J·cm^−2^ light 15 min or 24 h after the injection, as indicated. Data are expressed as the relative tumor growth (% of tumor volume on the day of light exposure) ± SEM in each group (*n* = 8, except in the group treated with free verteporfin and exposed to light 15 min after IV injection of verteporfin, in which 5 mice died during the experiment, as illustrated by skulls). Dotted red lines represent the tumor volume of each mouse from the group exposed to 50 J·cm^−2^ light, 15 min after free verteporfin injection. (**d**) Mice were intravenously injected with 1× PBS (control), or NLC-verteporfin (verteporfin 8 mg·kg^−1^), and exposed to 200 J·cm^−2^ light, 24 h after the injection as indicated. Data are expressed as the relative tumor growth (% of tumor volume at the day of light exposure) ± SEM in each group (*n* = 8). *p* = 0.0003 ANOVA with Tukey post-hoc tests (**, ***, significantly different from the control group). (**e**) Hematoxylin and eosin staining on frozen tumor sections, Scale bar: 50 µm. Representative images in each group are shown.

**Figure 6 cancers-11-01760-f006:**
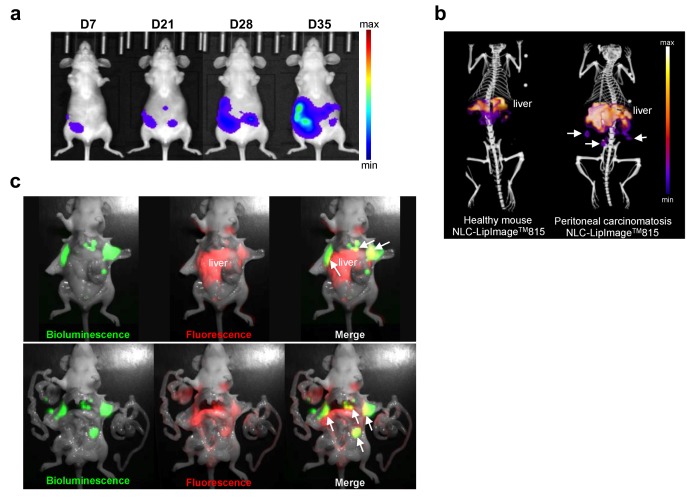
NLC-LipImage accumulated in ovarian tumor nodules after intravenous administration. (**a**) Orthotopic murine model of peritoneal carcinomatosis from ovarian cancer. Mice were inoculated with SKOV3-Luc cells into one ovary. Tumor growth and dissemination of tumor cells in the peritoneal cavity were followed by in vivo bioluminescence imaging over time. (**b**) Mice with peritoneal carcinomatosis or healthy mice were injected intravenously with NLC-LipImage^TM^-815 for 24 h, and imaged with fluorescence tomography/micro-computed tomography (microCT). Arrows show NLC-LipImage fluorescence in tumor nodules. (**c**) Mice with established peritoneal carcinomatosis were injected intravenously with NLC-LipImage, and were submitted to intraoperative fluorescence in combination with bioluminescence after 24 h. Representative images of one operated mouse (upper panel), and with extended organs exposition after liver removal (lower panel) are shown. Bioluminescence showed SKOV3 cells in the peritoneal cavity (in green) and 2D-fluorescence imaging showed NLC-LipImage location (in red). Arrows show NLC-LipImage and SKOV3 tumor nodules signals colocation (in yellow).

**Table 1 cancers-11-01760-t001:** Physico-chemical and encapsulation properties of nanostructured lipid carriers (NLC).

Properties	NLC-Verteporfin		NLC-LipImage^TM^815
	T0	T0 + 3 Months	
Hydrodynamic diameter (nm) ^a^	47.9 ± 1.0	54.4 ± 0.6	46.1 ± 0.7
Polydispersity index ^a^	0.12 ± 0.02	0.18 ± 0.01	0.13 ± 0.01
Zeta potential (mV) ^a^	−3.7 ± 0.9	−2.0 ± 1.3	−4.2 ± 4.3
Verteporfin (µg/mL) ^b^	1026.2 ± 15.6	1037.5 ± 0.7	/
Verteporfin concentration (µM) ^b^	1428	1443	/
LipiImage^TM^815 concentration (µM)	/	/	302

^a^ Dynamic Light Scattering DLC; ^b^ High Performance Liquid Chromatography HPLC.

**Table 2 cancers-11-01760-t002:** Sensitivity of ovarian cancer cells to verteporfin and NLC-verteporfin.

Cell Lines	Culture Conditions		Verteporfin IC_50_ (nmol·L^−1^)	NLC-Verteporfin IC_50_ (nmol·L^−1^)
SKOV3	2D	2 h	17.8 ± 0.9	7.3 ± 0.4
		24 h	23.8 ± 0.9	8.8 ± 0.5
	3D	2 h	41.2 ± 4.2	29.3 ± 7.1
		24 h	9.3 ± 0.7	9.7 ± 1.1
OVCAR3	2D	2 h ^#^	117.5 ± 14.4	116.4 ± 22.7
		24 h	94.9 ± 15.4	97.5 ± 17.4
	3D	2 h	28.9 ± 0.8	36.6 ± 3.2
		24 h	9.7 ± 1.0	5.3 ± 0.5

The drug concentrations required to inhibit cell growth by 50% (IC_50_) at 2 h or 24 h in SKOV3 or OVCAR3 cells cultured in monolayer (2D) or spheroids (3D) and after light exposure (690 nm; 10 J·cm^-2^). Data represent the mean ± SD (SKOV3, *n* = 3; OVCAR3, *n* = 4, except ^#^
*n* = 2). Each independent experiment was performed in triplicate.

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
