# Peer review of "Verteporfin-Loaded Lipid Nanoparticles Improve Ovarian Cancer Photodynamic Therapy In Vitro and In Vivo"

_cancers, 2019, doi:10.3390/cancers11111760_

Round 1

Reviewer 1 Report

The work from Thierry MICHY and co-workers concerns the preparation of a lipid-based nanoparticle formulation for delivering the photosensitizer Verteporfin, for the treatment of ovarian cancer. The work deals with both the in vitro and the in vivo characterizations.

As a general consideration, the work is well designed. The results are adequately presented and discussed and, most importantly, the conclusions are results driven. The strength of this work also relies on the in vivo characterization of the nanoparticles formulation, highlighting the complexity of the in vivo situation. Moreover, the use of spheroids as an in vitro tool is for sure a plus, compared to the 2D cell cultures.

All these aspects make this work original and deserving to be published, following minor comments/modifications:

Lines 68-74: this part of the introduction presents the results of the research. I feel to suggest to the authors to delete the results here, leaving space to describe the outlines followed in the work. Line 81: How was the 943 µg of drug for 100 mg of lipids calculated? Please, report this information in the methods section. Line 82: Under with conditions the small stability study was performed? Light or dark? Liquid or dried state? Table 1: I suggest to the authors to replace the name “droplet size distribution” with “polydispersity index” as previously indicated. Table 1: Regarding the Z-potential of NLC-LipImage™ 815, the dispersity of the measure is quite high. Can the authors comment this aspect and give an explanation? Line 119: why in Fig. 3 only the results from SKOV3 cells were reported? In other words, why the results of OVCAR3 are in supplementary file? Furthermore, which could be the reason why the SKOV3 and the OVCAR3 cells had a different sensibility to Verteporfin and to NLC-verteporfin? Why were the results on the Verteporfin and NLC-verteporfin sensitivity for IGROV1 cell line nor reported in Table 2? Line271: the encapsulation efficiency of verteporfin was not calculated and reported on the results (Table 1), where only the drug content is presented. I suggest to the authors to calculate it as efficiency respect to the initial amount of drug used for the formulation preparation. Line 273: did the authors check the production of loaded NLC of large volumes? Line 287-288: this sentence is not clear. Can the authors re-phrase it? Line 378: why the formulation was filtered with a 0.22 µm filter? Did the authors check the lipid or the drug retention by the filter? Line 379: how were the lipid nanoparticles freeze-dried? Under which conditions? Did the authors check that the drying method do not alter the properties of the formulation? Line 397: did the authors used NLC as liquid formulation for Verteporfin quantification? If so, how was the amount of drug normalized for the amount of lipid, as presented in Table 1? Can the authors explain this aspect, as usually dried formulations are employed? Paragraph 4.6: why were used three cell lines but most of the results in the paper are related to SKOV3 cells? For the flow cytometry experiments, did the authors used fluorescent formulations? How was it prepared? Line 428: were the cells maintained under dark or light conditions? Few typo errors are present throughout the manuscript (e.g. line 418, 423, 442..).

Author Response

The work from Thierry MICHY and co-workers concerns the preparation of a lipid-based nanoparticle formulation for delivering the photosensitizer Verteporfin, for the treatment of ovarian cancer. The work deals with both the in vitro and the in vivo characterizations.
As a general consideration, the work is well designed. The results are adequately presented and discussed and, most importantly, the conclusions are results driven. The strength of this work also relies on the in vivo characterization of the nanoparticles formulation, highlighting the complexity of the in vivo situation. Moreover, the use of spheroids as an in vitro tool is for sure a plus, compared to the 2D cell cultures. All these aspects make this work original and deserving to be published, following minor comments/modifications:

Lines 68-74: this part of the introduction presents the results of the research. I feel to suggest to the authors to delete the results here, leaving space to describe the outlines followed in the work.
As suggested by the reviewer, we have changed the text as follows lines 69-74: “The cell-uptake and phototoxicity of NLC-verteporfin were first assessed in vitro using 2D-monolayers and 3D-spheroids of human ovarian cancer cells. Biodistribution and pharmacokinetic studies of NLC-verteporfin were then evaluated in vivo after intravenous injection. Finally, the PDT efficacy of NLC-verteporfin was compared to free verteporfin in mice with ovarian cancer tumors. This study provides promising therapeutic prospects for PDT in ovarian cancer.”

Line 81: How was the 943 µg of drug for 100 mg of lipids calculated? Please, report this information in the methods section.
The analytical method for the analysis of the verteporfin payload of NLC is described in the methods section. Practically, after formulation and dialysis, the NLC concentration is first measured by lyophilizing aliquots of 100 µL, then weighting the dried aliquots. This experiment gives access to the particle concentration (in mg/mL), as stated in section 4.2.
Then, as described in section 4.4 (lines 427-431), 100 µL of NLC dispersion (with known weight concentration) are added to 100 µL of methanol. Samples are centrifuged to pellet lipids after precipitation, and the supernatant is taken for analysis. The lipid pellet is rinsed 3 times more with 100 µL methanol. The total supernatant, adjusted to 500 µL, is centrifuged once more to remove any remaining lipids and injected to HPLC analysis for verteporfin quantitation.
We have specified it the text in section 4.4 as follows line 427: “For analysis of the verteporfin payload in NLC, nanoparticles were disassembled.” and line 430: “The total supernatant, adjusted to 500 µL, was centrifuged once more to remove any remaining lipids and injected to HPLC analysis for verteporfin quantitation.”

Line 82: Under with conditions the small stability study was performed? Light or dark? Liquid or dried state?
The stability study was performed in dark in liquid state.
It is now specified in the Results (lines 81-83): “This formulation was stable for at least 3 months (size variation <10%, polydispersity index <0.2), when stored at 4°C in dark in water (concentration of ̴ 100 mg.mL-1 of lipids) (Table 1).”, and in the Methods (lines 411-412): “Physical stability was investigated by DLS measurements over 3 months with samples stored at 4°C in dark.”

Table 1: I suggest to the authors to replace the name “droplet size distribution” with “polydispersity index” as previously indicated.
It has been replaced.

Table 1: Regarding the Z-potential of NLC-LipImage™ 815, the dispersity of the measure is quite high. Can the authors comment this aspect and give an explanation?
The Z-potential error seems quite high in comparison to the mean absolute value of Z-potential because NLC are close to neutrality. Nanoparticles are usually considered as positive (resp. negative) when Z-potential is above +20 (resp. below -20) mV. Z-potential deviations of +/- 2-3 mV are usual. It is true that for this particular NLC-LipImageTM815 batch it was a bit high.

Line 119: why in Fig. 3 only the results from SKOV3 cells were reported? In other words, why the results of OVCAR3 are in supplementary file? Furthermore, which could be the reason why the SKOV3 and the OVCAR3 cells had a different sensibility to Verteporfin and to NLC-verteporfin? Why were the results on the Verteporfin and NLC-verteporfin sensitivity for IGROV1 cell line nor reported in Table 2?
As suggested, we have now moved the results on OVCAR3 cells and spheroids from supplementary file to the new figure 3. We have also added experiments on OVCAR3 spheroids and light exposure after 2h of treatment.
SKOV3 and OVCAR3 cell lines are both ovarian carcinoma cells. According to the NIH, OVCAR3 is an appropriate model system in which to study drug resistance in ovarian cancer, suggesting that these cells resists to drugs. We showed here that OVCAR3 cells also resisted to PDT agents, although we do not know why. We have added in the discussion (line 290): “The mechanism by which the OVCAR3 cells resisted to PDT in 2D culture but not in spheroids is still unknown.”
We analyzed verteporfin and NLC-verteporfin binding using flow cytometry in the three cell lines. Next, we only used SKOV3 and OVCAR3 cells because we failed to culture IGROV1 cells in 3D-spheroids. We feel that our in vitro studies with 2 cell lines cultured in 2D-monolayer and 3D-spheroids are sufficiently convincing to show the phototoxicity of verteporfin and NLC-verteporfin.

Line 271: the encapsulation efficiency of verteporfin was not calculated and reported on the results (Table 1), where only the drug content is presented. I suggest to the authors to calculate it as efficiency respect to the initial amount of drug used for the formulation preparation.
The verteporfin encapsulation yield in NLC was quantitative (>95%), as reported line 80.
Other lipophilic fluorescent dyes have also been reported to encapsulate quantitatively in NLC (J. Gravier, F. Navarro, T. Delmas, F. Mittler, A.C. Couffin, F. Vinet, I. Texier. Lipidots: a competitive organic alternative to quantum dots for in vivo imaging, Journal of Biomedical Optics 2011, 16(9) 096013).
We have added in the Discussion (line 278): “NLC showed high verteporfin encapsulation efficiency and stability for several months, as previously shown [21].” and the reference Gravier et al, 2011 (ref 21 in the manuscript).

Line 273: did the authors check the production of loaded NLC of large volumes?
We have implemented in our lab the production of NLC at large scale using High Pressure Homogenization (HPH), followed by Tangential Flow Filtration (TFF) for particle purification. We usually process about 400 mL of formulation by HPH, and 75 mL by TFF (about 15 g of particles). Results are presently submitted for publication and are under review. However, for the study presented here, lab-scale production is sufficient (about 700 mg of particles are obtained for a 2 mL batch) and avoids engaging large quantity of active molecule verteporfin.

Line 287-288: this sentence is not clear. Can the authors re-phrase it?
We have modified the sentences line 291 as follows: “These results highlighted that the toxicity of drugs significantly varied depending on whether it is assessed in spheroids or in monolayer cell cultures, and this thus underlined the importance of using spheroids before moving forward in vivo experiments in mice”, and line 296 as follows: “Taking into account cellular and microenvironment heterogeneity within tumors is particularly important to enhance the accumulation of nanoparticles in the tumor.”

Line 378: why the formulation was filtered with a 0.22 µm filter? Did the authors check the lipid or the drug retention by the filter?
The formulation is filtered on 0.22 µm filter after purification and before conditioning to perform sterilizing filtration. We have specified it line 406:” Prior to characterization and injection, NLC dispersions were filtered through a 0.22 µm cellulose Millipore membrane for sterilization.”
Analysis of lipid retention onto the filter was previously performed (Mathieu Varache, Mathieu Ciancone, Anne-Claude Couffin, “Development and validation of a novel UPLC-ELSD method for the assessment of lipid composition of nanomedicine formulation”, Int J Pharm 2019, 566, 11- 23) and quantified to be less than 5% for this particular process step. Concerning drug retention on the filter, since drug recovery is >95%, it can be assumed to be negligible. To note also that verteporfin is highly colored, and that no visual indication of dye retention on the filter was observed with naked eye.

Line 379: how were the lipid nanoparticles freeze-dried? Under which conditions? Did the authors check that the drying method do not alter the properties of the formulation?
The objective of freeze drying is here just to obtain the nanoparticle concentration in the formulation (in mg/mL). Freeze-drying is performed on usual chemistry lab freeze dryer (FreeZone, Labconco): 100 µL of particle suspension are pipetted into a tared microtube, the microtube is inserted a few minutes in liquid nitrogen, then set into the lyophilizer for 24h, the microtube is weighted at the end to determine the weight of freeze dried dispersion. For this measurement, we do not need to keep the nanoparticle intact. In fact, we have previously observed that freeze-drying alters particle structure (particle size increases after reconstitution). We have also developed freeze-drying protocol limiting particle size increase after reconstitution, based on the addition of trehalose as cryoprotectant. However, it is not necessary here to use such complex protocol, since particle dispersion is very stable in water (for more than 3 months). Storage of NLC formulation is therefore preferred in liquid suspension, with storage at 4°C in dark.
It has been now specified line 407: “NLC formulation was stored in liquid suspension at 4°C in dark.”

Line 397: did the authors used NLC as liquid formulation for Verteporfin quantification? If so, how was the amount of drug normalized for the amount of lipid, as presented in Table 1? Can the authors explain this aspect, as usually dried formulations are employed?
We used liquid formulation of verteporfin for storage. For quantification, we indeed measured the amount of lipids by freeze-drying. As explained above, the quantification protocole is the following: Practically, after formulation and dialysis, the NLC concentration is first measured by lyophilizing aliquots of 100 µL, then weighting the dried aliquots. This experiment gives access to the particle concentration (in mg/mL), as stated in section 4.2 (line 406). Then, as described in section 4.4 (lines 427-431), 100 µL of NLC dispersion (with known weight concentration) are added to 100 µL of methanol. Samples are centrifuged to pellet lipids after precipitation, and the supernatant is taken for analysis. The lipid pellet is rinsed 3 times more with 100 µL methanol. The total supernatant, adjusted to 500 µL, is centrifuged once more to remove any remaining lipids and injected to HPLC analysis for verteporfin quantitation.

Paragraph 4.6: why were used three cell lines but most of the results in the paper are related to SKOV3 cells?
We used three cell lines for verteporfin and NLC-verteporfin binding analyzes using flow cytometry. Next, we only used SKOV3 and OVCAR3 cells because we failed to culture IGROV1 cells in 3D-spheroids. We feel that our in vitro studies with 2 cell lines cultured in 2D-monolayer and 3D-spheroids are sufficiently convincing to show the phototoxicity of verteporfin and NLC-verteporfin. We have now moved the results on OVCAR3 cells and spheroids from supplementary file to the new figure 3.
For in vivo experiment, SKOV3 cells were used in two different models: subcutaneous tumors and orthotopic tumors.

For the flow cytometry experiments, did the authors used fluorescent formulations? How was it prepared?
We followed the intrinsic fluorescence of verteporfin, which have fluorescence emission signal at 690 nm after excitation at 420 nm (see Figure S1b). This is described lines 89-91. We have also added the following sentence in the methods (4.7) line 484: “Fluorescence emission of verteporfin was analyzed using flow cytometry LSRII and FCS Express software (BD Biosciences, San Jose, USA).”

Line 428: were the cells maintained under dark or light conditions?
The cells were maintained in the dark for the flow cytometry experiments. We only wanted to follow the binding and internalization of verteporfin or NLC-verteporfin, and not to activate the verteporfin specifically at 689 nm. We have added (line 460): “in the dark”.

Few typo errors are present throughout the manuscript (e.g. line 418, 423, 442..).
The manuscript has been carefully re-readed to remove typo errors.

Reviewer 2 Report

Photodynamic therapy is not a new approach to treat cancer, indeed there are many papers in the literature reporting this together with the encapsulation of several photosensitizer molecules. However, it represents an interesting topic.
The present manuscript, describing an elegant experimental work, is an enjoyable paper to read. This reviewer is also aware about the difficulty to perform some of the experiments shown here.

General comments.

1.- Drug encapsulation promotes relevant advantages in comparison with the free drug. This is clearly stated in the literature as occur here with the encapsulation of the photosensitizer.

2.- Regarding the efficacy, the authors claim that the tumor area was not entirely illuminated leading to a regrowth. However, even in this case, it should be good to explore the optimization of the dose regimen of NLC in order to achieve long-term antitumor efficacy. Indeed, the combination with surgery, may occur to have an area more extensive than the probe for the light illumination, supporting then, the necessity to find adequate regimens.

3.- Orthotopic model is more closer than subcutaneous, then it would be interesting to test the NLC-verteporfin activity in this model, once the results have demonstrated tumor accumulation of NLC-LipImage.

4.-Since PDT is able to induce not only hypoxia but also immune response, may be possible to investigate the immunogenic cell death and therefore, the immune system activation in this therapy? Do you know the role of this, immune response, in the current treatment?, please give any comment

5.-Verteporfin shows a pk characteristics that limit its efficacy, at least for tumor treatment, but could be possible to find the dose and, specifically, the time for light exposure in order to reduce the toxicity? or is this impossible to reach?

6.- Table 2 shows the IC50 values for free and encapsulated therapeutic agent for two ovarian cell lines. It is curious that for short exposure times the IC50 72h later is lower than for longer exposure times in 2D culture for SKOV3 cells and not for OVCAR3, please comment this difference.

7.- Figure 4 represents different plots regarding the kinetics of NLC. It is clear that the PK of free drug limits the distribution to the tumor and thereby, the efficacy/toxicity. Interesting Figure 5, panel b where is observed 4 responder vs. 1 non-responder, What is the interindividual variability for this type of therapies?
I suggest to remove the mean values for this group in this figure (5b). 

8.-According to the current results, what may be the most relevant factor to achieve adequate efficacy/toxicity ratio for this type of therapeutic strategy? Is time-dependent, dose-dependent or light-potency dependent?

9.-Finally, the high accumulation of NLC in liver might be used for the hepatocarcinoma treatment?

Suggestion:
It would be good to present a graph with the working flow, because there are many experiments involved in the goal.

Author Response

Comments and Suggestions for Authors

Photodynamic therapy is not a new approach to treat cancer, indeed there are many papers in the literature reporting this together with the encapsulation of several photosensitizer molecules. However, it represents an interesting topic. The present manuscript, describing an elegant experimental work, is an enjoyable paper to read. This reviewer is also aware about the difficulty to perform some of the experiments shown here.

General comments.

1.- Drug encapsulation promotes relevant advantages in comparison with the free drug. This is clearly stated in the literature as occur here with the encapsulation of the photosensitizer.
We are grateful to the reviewer to recognize the relevance of our work.

2.- Regarding the efficacy, the authors claim that the tumor area was not entirely illuminated leading to a regrowth. However, even in this case, it should be good to explore the optimization of the dose regimen of NLC in order to achieve long-term antitumor efficacy. Indeed, the combination with surgery, may occur to have an area more extensive than the probe for the light illumination, supporting then, the necessity to find adequate regimens.
We agree with this comment. To complete our discussion, we have added line 355: “Altogether, these data suggested that the dose regimen of NLC-verteporfin have to be optimized to achieve long-term antitumor efficacy.”

3.- Orthotopic model is more closer than subcutaneous, then it would be interesting to test the NLC-verteporfin activity in this model, once the results have demonstrated tumor accumulation of NLC-LipImage.
Accordingly, our results showing that NLC-LipImage accumulated in small tumor nodules in peritoneal carcinomatosis strongly suggested that treating peritoneal metastases by PDT should be investigated. However, we feel that such experiment should be performed in another study, along with survival assessment after the surgery with and without combinatory PDT.
We have added in the discussion lines 332-335: “Our study is the first, to our knowledge, to show PDT efficacy in subcutaneous ovarian tumors after single intravenous administration of NLC-verteporfin and laser light exposure. These results should be reinforced by using orthotopic model of peritoneal carcinomatosis from ovarian cancer.”

4.-Since PDT is able to induce not only hypoxia but also immune response, may be possible to investigate the immunogenic cell death and therefore, the immune system activation in this therapy? Do you know the role of this, immune response, in the current treatment? please give any comment
Antitumor effects of PDT result from the combination of three independent mechanisms involving direct cytotoxicity to tumor cells, destruction of tumor vasculature and induction of inflammatory and immune response. PDT-mediated inflammatory reaction is accompanied by tumor infiltration of the leukocytes, enhanced production of pro-inflammatory factors and cytokines, leading to systemic and specific antitumor immune response. In addition, PDT can be combined to cancer immunotherapies, to develop synergistic methods for cancer therapies. Exploring the role of PDT in cancer immunotherapies is a very interesting and exciting question, but we believe that this is outside the purpose of this article. Moreover, as far as we know there is no immunocompetent murine model of peritoneal carcinomatosis from ovarian origin that we could use for in vivo PDT experiments.
We have added in the discussion lines 355-358: “Altogether, these data suggested that the dose regimen of NLC-verteporfin have to be optimized to achieve long-term antitumor efficacy. Furthermore, the effect of PDT on induction of inflammatory and immune response, as well as its combination with cancer immunotherapies might be investigated to enhance the PDT response [5, 6, 13, 34].”

5.-Verteporfin shows a pk characteristics that limit its efficacy, at least for tumor treatment, but could be possible to find the dose and, specifically, the time for light exposure in order to reduce the toxicity? or is this impossible to reach?
Free verteporfin at the high dose we used had strong toxicity when the tumors were illuminated 15 min after verteporfin intravenous administration, and no effect when illuminated 24h after verteporfin intravenous administration (the verteporfin was already eliminated). In addition, free verteporfin has no specificity for the tumor. We supposed that the therapeutic efficacy of free verteporfin was mainly due to its vascular effect. The strong adverse effect that we observed with free verteporfin was mainly due to circulating verteporfin. We think that reducing the toxicity of circulating free verteporfin will be very difficult. Using NLC-verteporfin allowed enhancing the tumor uptake of verteporfin by EPR effect, having its therapeutic efficacy on tumor (and not on vasculature).
To complete our discussion, we have added line 313: “Altogether, this strongly suggested the need to formulate verteporfin to improve its tumor specificity and to decrease its side-effects.”

6.- Table 2 shows the IC50 values for free and encapsulated therapeutic agent for two ovarian cell lines. It is curious that for short exposure times the IC50 72h later is lower than for longer exposure times in 2D culture for SKOV3 cells and not for OVCAR3, please comment this difference.
We have no explanation for this event. However, we believe that this difference is not significant, as obviously observed in Fig 3. The IC50 reported in Table 2 are interpolated from the dose-response curves, and represent the best-fit values.

7.- Figure 4 represents different plots regarding the kinetics of NLC. It is clear that the PK of free drug limits the distribution to the tumor and thereby, the efficacy/toxicity. Interesting Figure 5, panel b where is observed 4 responder vs. 1 non-responder, What is the interindividual variability for this type of therapies? I suggest to remove the mean values for this group in this figure (5b). 
The individual responses were different from each other in the group treated with free verteporfin and exposed to light 15 min after IV. We agree that the individual responses were more relevant than the mean values for this group. This variability highlights the need to have sufficiently large groups of animals (at least n=8) to take into account this variability and obtain significant results.
As suggested, we have removed the mean values for the group treated with free verteporfin and exposed to light 15 min after IV in Figure 5b.

8.-According to the current results, what may be the most relevant factor to achieve adequate efficacy/toxicity ratio for this type of therapeutic strategy? Is time-dependent, dose-dependent or light-potency dependent?
This is a very interesting question. According to our results showing that injection of NLC-verteporfin at 8 mg/kg of verteporfin combined with 200 J/cm2 light exposure had a better efficacy than injection of NLC-verteporfin at 2 mg/kg of verteporfin combined with 50 J/cm2 light exposure, we suppose that both time and light fluency are important. In addition, the interval between drug administration and light exposure is to optimize. The accumulation of the photosensitizer in the tumor, thanks to a relevant drug delivery system, is of course also determining. Further studies are needed to determine which factor is the most relevant.
We have added in the Discussion (lines 337-339): “Here, we demonstrated that tumor growth inhibition was related to the combination of NLC-verteporfin dose (8 mg.kg-1 of verteporfin) and laser light fluency (690 nm and 200 J.cm-2), these factors yet needing further optimization.”

9.-Finally, the high accumulation of NLC in liver might be used for the hepatocarcinoma treatment?
This suggestion seems obvious since NLC strongly accumulated in the liver. However we believe that this is not suitable for hepatocarcinoma treatment, because NLC accumulate mainly in Küppfer cells and not in hepatocytes. In the hepatocarcinoma context, the NLC would thus accumulate in the liver Küppfer cells and not in the liver tumor cells.

Suggestion: It would be good to present a graph with the working flow, because there are many experiments involved in the goal.
We have changed the end of the Introduction to better describe the outlines followed in the work as follows (lines 69-74): “The cell-uptake and phototoxicity of NLC-verteporfin were first assessed in vitro using 2D-monolayers and 3D-spheroids of human ovarian cancer cells. Biodistribution and pharmacokinetic studies of NLC-verteporfin were then evaluated in vivo after intravenous injection. Finally, the PDT efficacy of NLC-verteporfin was compared to free verteporfin in mice with ovarian cancer tumors. This study provides promising therapeutic prospects for PDT in ovarian cancer.”

Reviewer 3 Report

The premise of this study was to demonstrate that encapsulation of the photosensitizer, verteporfin, within lipid particles can be used as a method to promote drug/lipid carrier accumulation within tumors for photodynamic therapy. The authors demonstrate uptake verteporfin by ovarian cancer cell lines and tumor xenografts. They also provide some limited PK data show that mice treated with NLC-verteporfin followed by photodynamic therapy slows tumor growth. Additionally, they demonstrate that their approach reduces toxicity in mice.

Thank you for the opportunity to review your work. I enjoyed reading your paper. This paper appears to be solid and well-written. The experiments seem to have the correct controls and have been repeated multiple times to provide convincing results and the appropriate statistical analysis. I do think there are a few points that should be addressed.

Major points

Figure 5e shows H&E staining of frozen tumor sections from treated mice. As worded, “In addition, necrotic areas were observed in tumors treated with NLC-verteporfin and laser light” it isn’t clear if necrotic areas were observed in tumors from the NLC-verteporfin/laser-treated mice and tumors from NLC-verteporfin mice or only the NLC-verteporfin/laser-treated mice. Please clarify. It also looks as if there may be necrotic areas in tumors from mice injected with NCL-verteporfin, but not exposed to the laser. The authors should clarify and provide higher magnification images. How many fields were viewed and from how many different tumors in each treatment group? Were the sections viewed by a pathologist? If necrotic areas are present in tumors from the non-laser treated mice, this should be clearly pointed out in the text.

Based on the high accumulation of verteporfin in the liver (Fig. 4d and f), did the authors analyze the liver and other tissues for signs of toxicity/necrosis, especially since it appears that there may be a low level of necrosis in the tumor tissue? Is there any other previous information available regarding liver or other tissue toxicity based on pathological evaluation of organs?

Line 118: “NLC-verteporfin exposed to NIR light 117 significantly reduced the viability of SKOV3 and OVCAR3 cells cultured in monolayers”. No p-value is provided and there is no evidence of a statistical test. Please provide or remove the word “significant” from the text.

Table 2: No data is provided for OVACR-3 cells at 2 hours, as with the other lines. While inclusion of these data would not change the overall message or conclusions of the paper, it is odd that they are not included. Where these experiments not done?

There seems to be an issue with the curve fitting for Fig. 3a, 2-hour light treatment. Please check to make sure that the curve fitting was done correctly (10 nmol *L-1) data point. Legend for Fig 3b is not correct (10 J vs. light).

Fig 4f-label should read ug/g of “tissue” instead of “organ” since tumors are not organs.

Line 184: “When tumors were exposed to laser light 15 min after free verteporfin injection (2 mg.kg-1), their growth was immediately and strongly 184 reduced (Figure 5b).” Do the authors know if this was due to tumor rupture? It would seem a likely possibility given the observed toxicity. If possible, this should be addressed in the discussion. Also, the wording here should better describe the observation as “immediate” implies right after treatment. It appears that the size declined over a matter of days.

Line 209: (Figure S8a), suggesting vascular impairment resulting in tissue hypoxia in the treated tumors. Can the authors follow up this observation with staining for CD31 to assess vascular structure and vessel numbers? It would seem that these would be highly supportive data, and it would solidify the statement rather than leaving it to conjecture. 

Lines 348-351: “Actually, PDT has already been demonstrated to be easily combined with conventional surgical tumor resection to improve treatment outcome of peritoneal carcinomatosis [3]. Intraoperative PDT could be applied immediately following surgical tumor debulking and could treat residual peritoneal tumors in areas where surgical procedures pose a high risk of perioperative complications [35, 36].”

The authors need to provide a more realistic assessment of the field in this section. While in theory, PDT can be used during surgery, this is not a simple procedure for the surgeon or the patient. Therefore, the authors should remove the word “easily” in line 348. Also, toxicity has been observed in the patients in the references cited. This approach may be viable for low disease burden, but it is quite challenging for high disease burden. This is especially true in ovarian cancer where hundreds of metastatic sites may be present within the peritoneal cavity. While the authors do meet address one challenge with their work, the remaining challenge of how to use their approach effectively on patients, and whether the focus should be on patients with low metastatic vs. high metastatic burden should be addressed more carefully in the Discussion.

Author Response

Comments and Suggestions for Authors

The premise of this study was to demonstrate that encapsulation of the photosensitizer, verteporfin, within lipid particles can be used as a method to promote drug/lipid carrier accumulation within tumors for photodynamic therapy. The authors demonstrate uptake verteporfin by ovarian cancer cell lines and tumor xenografts. They also provide some limited PK data show that mice treated with NLC-verteporfin followed by photodynamic therapy slows tumor growth. Additionally, they demonstrate that their approach reduces toxicity in mice.

Thank you for the opportunity to review your work. I enjoyed reading your paper. This paper appears to be solid and well-written. The experiments seem to have the correct controls and have been repeated multiple times to provide convincing results and the appropriate statistical analysis.
We are very grateful to receive such encouraging comments.

I do think there are a few points that should be addressed.
Major points
Figure 5e shows H&E staining of frozen tumor sections from treated mice. As worded, “In addition, necrotic areas were observed in tumors treated with NLC-verteporfin and laser light” it isn’t clear if necrotic areas were observed in tumors from the NLC-verteporfin/laser-treated mice and tumors from NLC-verteporfin mice or only the NLC-verteporfin/laser-treated mice. Please clarify. It also looks as if there may be necrotic areas in tumors from mice injected with NCL-verteporfin, but not exposed to the laser. The authors should clarify and provide higher magnification images. How many fields were viewed and from how many different tumors in each treatment group? Were the sections viewed by a pathologist? If necrotic areas are present in tumors from the non-laser treated mice, this should be clearly pointed out in the text.
We have clarified the sentence as follows line 188: “In addition, necrotic areas were observed in tumors treated with NLC-verteporfin combined with laser light only.”
We have also increased the magnification of the images in figure 5e.
The whole tumor sections were scanned and observed (4 different tumors in each group, and 3 sections per tumor). The sections were not examined by a pathologist but by 2 researchers, who were blinded to the treatment groups. We did not observed necrotic areas in tumors from the NLC-verteporfin and dark group.
These points have been added in the methods (4.10.7, line 565) as follows: “Sections were observed under a Zeiss AxioImager M2 microscope by two researchers who were blinded to the treatment groups (4 different tumors per group).” and in the discussion line 341: “A specific quantification of tumor necrosis should be performed to confirm these observations.”

Based on the high accumulation of verteporfin in the liver (Fig. 4d and f), did the authors analyze the liver and other tissues for signs of toxicity/necrosis, especially since it appears that there may be a low level of necrosis in the tumor tissue? Is there any other previous information available regarding liver or other tissue toxicity based on pathological evaluation of organs?
Macroscopic observation of liver and other organs of mice did not show any evidence of damage. We also analyzed liver sections (4 livers from each group and 3 sections per liver) stained with hematoxylin and eosin (HE), without finding any signs of necrosis or damage. This suggested that NLC-verteporfin did not impair liver function despite their strong hepatic accumulation.
We have added representative images of liver sections from each group in the new supplementary figure S8. We have now mentioned it in the results (lines 189-191) as follows: “Macroscopic observation of the main organs did not show evidence of damage. In addition, liver sections did not show any evidence of necrosis or damage (Figure S8).”, and in the Methods (line 564): “Tumors and livers were frozen and sections of a 7-μm thickness were stained with hematoxylin and eosin (HE).”
In the literature, liver toxicity has been previously evaluated for dye loaded NLC (particles with the same composition except for the payload) through the quantitation of different enzymes as well as tissue histology. Studies were performed in rats (unpublished) and in dogs (D. Sayag, Q. Cabon, I. Texier, F. P. Navarro, R. Boisgard, D.Virieux-Watrelot, C. Carozzo, F. Ponce. Phase-0/phase-I study of dye-loaded lipid nanoparticles for near-infrared fluorescence imaging in healthy dogs, European Journal of Pharmaceutics and Biopharmaceutics 2016, 100, 85-93). All enzyme levels were normal for 7 days of monitoring after IV injection in dogs (2 weeks in rat). Organ histology has indicated no liver or other organ toxicity, even at relatively high dose of particles (ie about 40 mg/kg in dogs). Our data on liver sections suggested that NLC-verteporfin did not induce hepatic damage.
We have added in the discussion line 325: “In addition, no evidence of toxicity was observed in liver, despite the high hepatic accumulation of NLC, as it has been previously shown with dye-loaded NLC [30].” and the reference Sayag et al, 2016 (ref 30 in the manuscript).

Line 118: “NLC-verteporfin exposed to NIR light 117 significantly reduced the viability of SKOV3 and OVCAR3 cells cultured in monolayers”. No p-value is provided and there is no evidence of a statistical test. Please provide or remove the word “significant” from the text.
We have removed the word “significantly” from the text.

Table 2: No data is provided for OVACR-3 cells at 2 hours, as with the other lines. While inclusion of these data would not change the overall message or conclusions of the paper, it is odd that they are not included. Where these experiments not done?
We initially did not perform the cell viability experiments at 2h in OVCAR3 spheroids because we thought that NLC-verteporfin needed more time to enter and act in the spheroids. We have now perform these experiments and have added it in the new figure 3d and in the new Table 2.

There seems to be an issue with the curve fitting for Fig. 3a, 2-hour light treatment. Please check to make sure that the curve fitting was done correctly (10 nmol *L-1) data point. Legend for Fig 3b is not correct (10 J vs. light).
We have changed the curve fitting methods for the new figure 3. We have changed the IC50 in the new Table 2 accordingly. The legend for fig 3 has been corrected.

Fig 4f-label should read ug/g of “tissue” instead of “organ” since tumors are not organs.
We have corrected the fig 4f.

Line 184: “When tumors were exposed to laser light 15 min after free verteporfin injection (2 mg.kg-1), their growth was immediately and strongly reduced (Figure 5b).” Do the authors know if this was due to tumor rupture? It would seem a likely possibility given the observed toxicity. If possible, this should be addressed in the discussion. Also, the wording here should better describe the observation as “immediate” implies right after treatment. It appears that the size declined over a matter of days.
We postulated that the inhibition of tumor growth was related to free verteporfin action on tumor vasculature. Indeed, most experimental studies suggest that the main mechanism of action of verteporfin therapy after light activation is intravascular damage leading to thrombus formation and selective vascular occlusion (Battaglia 2016, ref 28 in the manuscript). Damaged endothelium is known to release procoagulant and vasoactive factors, resulting in platelet aggregation, fibrin clot formation and vasoconstriction. As singlet oxygen and reactive oxygen radicals are cytotoxic, verteporfin can also destroy tumor cells.
This is stated in the Discussion (lines 299-301): “the illumination of the retina is performed 15 min after intravenous injection of the photosensitizer, and the photodynamic reaction produces an anti-vascular effect that reduces disease progression”, and (lines 305-308): “In our work, illumination of subcutaneous SKOV3 ovarian tumors 15 min after intravenous injection of a high dose of verteporfin led to strong tumor regression, illustrating the strong phototoxic effect of verteporfin on vasculature,”.
Of course we cannot exclude that tumor growth inhibition following free verteporfin and light exposure at 15 min was due to tumor rupture. We have completed the discussion as follows line 309: “The inhibitory effect of verteporfin on tumor vasculature and/or tumor cells, and its relationship to side-effects, remain to be formally investigated.”
We have changed the word “immediately” (line 167) by “rapidly”.

Line 209: (Figure S8a), suggesting vascular impairment resulting in tissue hypoxia in the treated tumors. Can the authors follow up this observation with staining for CD31 to assess vascular structure and vessel numbers? It would seem that these would be highly supportive data, and it would solidify the statement rather than leaving it to conjecture.
Assessing and quantifying the vascular structure by CD31 staining in tumor sections would give strong data to support our photoacoustic imaging results. However we believe that it should be perform just after the monitoring of StO2 in the tumor, and not 2 weeks later, when tumors were excised at the end of the experiment.
The relationship between vascular impairment and tumor hypoxia has been well described in the context of head and neck tumor response to radiotherapy (Rich 2018, ref 37 in the manuscript), and in this study CD31 staining was performed just after photoacoustic imaging.
We have added this point in the Discussion lines 360-362 as follows: “The reduced levels of StO2 observed in tumors responding to the treatment suggested vascular impairment and tissue hypoxia, although this remains to be demonstrated by CD31 staining in tumors.”

Lines 348-351: “Actually, PDT has already been demonstrated to be easily combined with conventional surgical tumor resection to improve treatment outcome of peritoneal carcinomatosis [3]. Intraoperative PDT could be applied immediately following surgical tumor debulking and could treat residual peritoneal tumors in areas where surgical procedures pose a high risk of perioperative complications [35, 36].”
The authors need to provide a more realistic assessment of the field in this section. While in theory, PDT can be used during surgery, this is not a simple procedure for the surgeon or the patient. Therefore, the authors should remove the word “easily” in line 348. Also, toxicity has been observed in the patients in the references cited. This approach may be viable for low disease burden, but it is quite challenging for high disease burden. This is especially true in ovarian cancer where hundreds of metastatic sites may be present within the peritoneal cavity. While the authors do meet address one challenge with their work, the remaining challenge of how to use their approach effectively on patients, and whether the focus should be on patients with low metastatic vs. high metastatic burden should be addressed more carefully in the Discussion.
Accordingly with this comment, we have removed the word “easily”.
We have also modified the Discussion as follows (lines 371-373): “Actually, PDT has already been demonstrated to be combined with conventional surgical tumor resection to improve treatment outcome of peritoneal carcinomatosis despite a narrow therapeutic index [3].”, (lines 375-378): “Significant toxicity induced by intraperitoneal PDT has been shown in phase I and II clinical trials, owing to the heterogeneous population of patients, who had poor prognosis and failed to respond to first-line treatments, and to the non-specificity of photosensitizers for tumor cells [3].” and (lines 381-383): “Treating residual peritoneal metastases by PDT after cytoreductive surgery should thus be further investigated depending on metastatic tumor burden.”

Reviewer 4 Report

The authors should address the following issues in order to make the paper suitable for publications:

1 It is not clear from Fig. 4 how the half-lives of verteporfin and NLC-verteporfin were estimated. Was it based on fluorescence intensity of dye-loaded NLC (LipImageTM-815)? Was the amount of verteporfin in blood estimated through HPLC? 

2. The therapeutic benefit of NLC-verteporfin is not evident from Fig. 4, since no nanoparticle (NLC) control was included. 

3. The authors should discuss on how verteporfin could be released from NLC after being internalized by target cancer cells.

Author Response

Comments and Suggestions for Authors
The authors should address the following issues in order to make the paper suitable for publications:

1 It is not clear from Fig. 4 how the half-lives of verteporfin and NLC-verteporfin were estimated. Was it based on fluorescence intensity of dye-loaded NLC (LipImageTM-815)? Was the amount of verteporfin in blood estimated through HPLC? 
In Fig 4a, the half-lives were estimated by fluorescence intensity measurements of LipImage for NLC-LipImage, and of verteporfin for NLC-verteporfin and free verteporfin. In addition, we also used HPLC to quantify the level of verteporfin in the blood plasma over time after injection of NLC-verteporfin and free verteporfin (see figure S4).
To be clearer, we have added in the methods section 4.10.2 (line 503): “Half-lives were measured from nonlinear regression fit analyses (two phase decay).” and in the legend of Fig 4a: “Fluorescence intensity measurements of verteporfin (for free verteporfin or NLC-verteporfin), or of LipImageTM-815 (for NLC-LipImage) were performed on blood plasma samples taken at different time points.” and in the results line 158: “HPLC, as another quantification method, also confirmed these results in tissues and blood plasma (Figure 4f and Figure S4).”

2. The therapeutic benefit of NLC-verteporfin is not evident from Fig. 4, since no nanoparticle (NLC) control was included. 
We did not include a group treated with empty NLC in Fig 5c and 5d. Instead we included a group treated with NLC-verteporfin in the dark. At the two concentrations studied (Fig 5c and 5d), NLC-verteporfin in the dark did not inhibit the tumor growth as compared to the control (untreated) group, demonstrated that NLC did not inhibit tumor growth. Since our purpose was to compare free verteporfin to encapsulated verteporfin, and according to the institutional guidelines for the use of experimental animals (especially the reduction rule: to minimize the number of animals used per study), we believe that it is not necessary to add a group of mice treated with empty NLC.
In addition, lipid nanoparticles have been widely used in the literature without showing any therapeutic effect independently of their drug loading (see Debele et al., 2015 (ref 13 in the manuscript); Laine et al., 2014 (ref 17); Hirsjarvi et al., 2013 (ref 18); Navarro et al., 2014 (ref 19); Sayag et al., 2016 (ref 30); Jacquart et al., 2013 (ref 40), for example…).

3. The authors should discuss on how verteporfin could be released from NLC after being internalized by target cancer cells.
We have previously observed by FRET-based imaging, which provides information on both the distribution of dyes as drug models and the integrity of lipid nanoparticles, a fast dissociation of NLC after their internalization, leading to complete particle dissociation 6h after the experiment started (J. Gravier, L. Sancey, S. Hirsjärvi, E. Rustique, C. Passirani, J.P. Benoît, J.L. Coll, I. Texier. FRET Imaging Approaches for in Vitro and in Vivo Characterization of Synthetic Lipid Nanoparticles. Molecular Pharmaceutics 2014, 11(9), 3133-3144). Based on these data, we postulate that NLC should rapidly be dissociated following internalization, thus releasing verteporfin.
We have added in the Discussion (lines 321-322): “These data and previous study [29] suggested that NLC were rapidly dissociated following their internalization in tumor cells, thus releasing verteporfin.”, and the reference Gravier et al., 2014 (ref 29) in the manuscript.

Round 2

Reviewer 2 Report

Any comment

Reviewer 4 Report

The authors have properly addressed the concerns.